# The Magnitude of NCD Risk Factors in Ethiopia: Meta-Analysis and Systematic Review of Evidence

**DOI:** 10.3390/ijerph19095316

**Published:** 2022-04-27

**Authors:** Fisaha Haile Tesfay, Kathryn Backholer, Christina Zorbas, Steven J. Bowe, Laura Alston, Catherine M. Bennett

**Affiliations:** 1Institute for Health Transformation, Deakin University, Geelong 3220, Australia; kathryn.backholer@deakin.edu.au (K.B.); c.zorbas@deakin.edu.au (C.Z.); laura.alston@deakin.edu.au (L.A.); catherine.bennett@deakin.edu.au (C.M.B.); 2College of Medicine and Public Health, Flinders University, Adelaide 5042, Australia; 3College of Medicine and Health Sciences, School of Public Health, Mekelle University, Mekelle P.O. Box 231, Ethiopia; 4Deakin Biostatistics Unit, Faculty of Health, Deakin University, Geelong 3220, Australia; s.bowe@deakin.edu.au

**Keywords:** NCD, risk factors, hypertension, overweight, Ethiopia

## Abstract

Background: Non-communicable Diseases (NCDs) and their risk factors are the leading contributors to morbidity and mortality globally, particularly in low- and middle-income countries including Ethiopia. To date, there has been no synthesis of the literature on the relative prevalence of NCD risk factors in Ethiopia. Methodology: We conducted a systematic review and meta-analysis of primary studies reporting on the prevalence of NCD risk factors in Ethiopia published in English from 2012 to July 2020. Pre-tested NCD search terms were applied to Medline, Embase, Scopus, CINAHL, and Global Health. Three reviewers screened and appraised the quality of the identified papers. Data extraction was conducted using a pilot tested proforma. Meta-analysis was conducted using Stata 16 and pooled prevalence estimated with associated 95% confidence intervals. Clinically heterogeneous studies that did not fulfil the eligibility criteria for meta-analysis were narratively synthesised. I^2^ was used to assess statistical heterogeneity. Results: 47 studies fulfilled the inclusion criteria and contributed 68 NCD risk factor prevalence estimates. Hypertension was the most frequently examined NCD risk factor, with a pooled prevalence of 21% (*n* = 27 studies). The pooled prevalence percentages for overweight and obesity were 19.2% and 10.3%, respectively (*n* = 7 studies each), with a combined prevalence of 26.8% (*n* = 1 study). It was not possible to pool the prevalence of alcohol consumption, smoking, metabolic disorders, or fruit consumption because of heterogeneity across studies. The prevalence of alcohol use, as reported from the included individual studies, ranged from 12.4% to 13.5% (*n* = 7 studies). More than 90% of participants met the WHO-recommended level of physical activity (*n* = 5 studies). The prevalence of smoking was highly variable, ranging between 0.8% and 38.6%, as was the prevalence of heavy alcohol drinking (12.4% to 21.1%, *n* = 6 studies) and metabolic syndrome (4.8% to 9.6%, *n* = 5 studies). Fruit consumption ranged from 1.5% up to the recommended level, but varied across geographic areas (*n* = 3 studies). Conclusion and recommendations: The prevalence of NCD risk factors in Ethiopia is relatively high. National NCD risk factor surveillance is required to inform the prioritisation of policies and interventions to reduce the NCD burden in Ethiopia.

## 1. Introduction

NCDs such as cardiovascular diseases (such as heart attacks and stroke), cancers, chronic respiratory diseases (such as chronic obstructive pulmonary disease and asthma), and diabetes are the leading contributors to premature death and disability around the world [1]. According to the World Health Organization (WHO), the major non-communicable disease (NCD) risk factors include behavioural and metabolic risk factors [1]. Behavioural risk factors comprise tobacco use, physical inactivity, excessive alcohol use, and dietary risks. Metabolic risk factors include raised blood pressure, overweight and obesity, hyperglycaemia, and hyperlipidaemia [2]. Various factors such as globalization, urbanization, and socioeconomic characteristics have been reported to drive widespread and changing population level behavioural and metabolic NCD risk factor profiles [3,4,5,6]. 

In terms of behavioural factors, tobacco (including second-hand smoke) contributes to 7.2 million deaths per year, globally [1]. Insufficient physical activity is linked to 1.6 million deaths [1], with 31.1% of adults being physically inactive [1]. Alcohol consumption contributes to 1.8 million deaths and 52 million Disability-Adjusted Life Years (DALYs) [7]. Low fruit and vegetable consumption contributes to 31% of coronary heart disease and 11% of ischemic stroke [8]. A diet high in ultra-processed foods and drinks contributes to 40% of NCD-related deaths globally [9,10,11,12]. With respect to metabolic risk factors, 18 million deaths are caused by high blood pressure, high body mass index, hyperglycaemia, and high total cholesterol annually [12].

In low- and middle-income countries (LMICs), NCDs have been associated with smoking, increased BMI, high blood pressure, low physical activity, and low fruit and vegetable consumption [5,13,14]. Similar to high-income countries, evidence from LMICs suggests that a low socioeconomic position is associated with smoking and excessive alcohol consumption [15]. Furthermore, groups with low socioeconomic positions in LMICs are less likely to consume fruits and vegetables, while high socioeconomic groups are more likely to be physically inactive, and consume more fats, salts, and processed food [15]. 

As in many other low-income regions, the magnitude and distribution of NCD risk factors in sub-Saharan Africa is increasing at an alarming rate [6]. For example, all age DALYS increased by 67% between 1990 and 2017 [16]. Social, cultural, and economic factors have driven this rapid increase in NCDs and their risk factors [17]. From a sociocultural perspective, before the rollout of antiretroviral therapy (ART), HIV was associated with weight loss, and large body size was perceived as a sign of good health due to the absence of HIV [6,16,17,18]. Similarly, middle class population groups are expanding in sub-Saharan Africa, and this increasing wealth is associated with the adoption of behavioural NCD risk factors. Furthermore, whilst globalization and urbanisation improves access to health care and food imports, it concurrently increases the exposure of the urban poor in some sub-Saharan African countries to foods and drinks that are highly processed, energy-dense, nutrient-poor, ubiquitous, and cheap [6]. 

A systematic review of the prevalence of various NCD risk factors such as hypertension, metabolic syndrome, obesity and overweight was completed in sub-Saharan Africa in 2013 [19,20,21]. The commonest NCD risk factors reported in sub-Saharan Africa were hypertension and overweight [22]. 

Comprehensive and routine NCD risk factor surveillance across populations has long been recognised as a vital component of NCD prevention and control by leading health organizations, including the WHO [22]. Nevertheless, such government-led actions have been largely inadequate and are not a priority in many countries, including in Ethiopia [23]. For example, the WHO STEPS survey on the prevalence of NCD risk factors in Ethiopia has not been completed since 2015 [24]. Sub-Saharan countries in Africa ultimately face numerous barriers to NCD risk factor monitoring, including limited resources and competing priorities such as infectious, nutritional, and maternal health problems [25,26].

In Ethiopia, considerable evidence on the prevalence of individual NCD risk factors such as hypertension, hyperglycaemia, and metabolic syndrome has been published [27,28,29]. Nevertheless, only one systematic review (conducted in 2011) has summarised the NCD risk factor profile of Ethiopia, identifying hypertension, hyperglycaemia (diabetes mellitus), tobacco use, harmful use of alcohol, being overweight/obese, and *khat* chewing as the commonest NCD risk factors [28]. Hence, this systematic review updates the previous systematic review; also, it has additionally determined the relative prevalence of the NCD risk factors. Identifying the most prevalent NCD risk factors will assist policy makers in prioritising NCD risk factors and focusing NCD prevention efforts to have the greatest impact on overall NCD burden. 

## 2. Methodology 

### 2.1. Protocol

A protocol for this review was registered in PROSPERO (registration number CRD42020196815). 

### 2.2. Context/Setting

This review focused on Ethiopia. Located in the horn of Africa, Ethiopia is the second most populous country in sub-Saharan Africa, with 96.5 million inhabitants as of July 2020 [30]. According to the 1994 constitution, Ethiopia is administratively structured into 10 regional states: Tigray, Afar, Amhara, Oromiya, Somali, Benishangul-Gumuz, Southern Nations Nationalities and Peoples (SNNP), Sidama, Gambela, and Harari. There are also two city council administrations: Addis Ababa and Dire Dawa [31]. Regional states in Ethiopia are the federal states of second level administrative structures from the top. 

The Ethiopian health system is structured into three tiers: (1) tertiary level health care (i.e., specialized hospital services with the capacity to treat 3.5–5.0 million people), (2) secondary level health care (i.e., general hospitals, which treat 1–1.5 million people), and (3) primary care services (e.g., primary hospitals, which serve 60,000–100,000 people) and health centres (which serve 40,000 people) [32]. Disease prevention is a primary focus of the primary health care services [33]. While primary health care is characteristically less successful in low-income countries, Ethiopia has performed well in improving healthcare access [34,35,36].

To date, the Integrated diseases surveillance system in Ethiopia mainly focuses on communicable diseases such as HIV, malaria, tuberculosis, and maternal death [37,38]. National surveys such as the WHO STEPS survey (2015) and the service availability survey (2016) have been used to document the prevalence of NCDs and some of their risk factors in Ethiopia [24,39]. 

### 2.3. Searches

A comprehensive search identified all peer-reviewed published studies that reported on the prevalence of NCD risk factors in Ethiopia from 1 January 2012 to 20 July 2020 as an update to the previous systematic review (from 1960 to 2011) [28]. Key search terms and keywords were identified through a preliminary search using the terms “prevalence” AND “non-communicable diseases” AND “Ethiopia”. Final key concepts and several corresponding search terms were then systematically applied (Table 1). The search was conducted across five electronic databases: Medline (Appendix B), Embase, Scopus, CINAHL, and Global Health. Grey literature was searched using Google search, Google Scholar, ProQuest, open grey, government websites (i.e., the Ethiopian Ministry of Health and Ethiopian Public Health Institute), and NGO websites such as the WHO. The reference lists of included studies were also searched using two terms: “non-communicable disease” AND “Ethiopia”. 

### 2.4. Inclusion Criteria and Article Selection

All search results were exported to Endnote and then to Covidence. Removal of duplicates, screening by title and abstract, and full-text screening were all completed in Covidence.

Articles were included if they met the following inclusion criteria: (1) primary quantitative study, (2) reported the prevalence of NCD risk factors in Ethiopia, (3) community-based study among adults. 

After removing the duplicates in Covidence, three of the study authors (FT, LA, and CZ) screened titles and abstracts against the inclusion criteria. Full-text screening was then completed by one of the same three authors. 

Studies with no clear objective, research question, and/or methodology were excluded. In addition, studies with a very small sample size (*n* < 50) were excluded because of the poor reliability of the estimates. 

Where discrepancies arose between the two reviewers, the third reviewer was invited to resolve the differences in article selection decisions. All reasons for the exclusion of studies were reported. 

### 2.5. Measurement of Outcome Variables 

The outcome variable was the prevalence of NCD risk factors overall and by type (e.g., hypertension, alcohol consumption, smoking, sedentary lifestyle, and other relevant risk factors). 

### 2.6. Assessment of Methodological Quality and Selection 

Quality assessment used the Newcastle Ottawa scale for cross-sectional studies [40,41,42]. A star rating is assigned to each study (with a maximum score of 10 stars). The parameters against which studies are assessed include: selection (maximum five stars), comparability (maximum three stars), and outcomes (two stars) [24]. The assessment tool has no cut-off points to grade studies as high, moderate, or low quality. Hence, a relative comparison of studies was made. Three authors (FT, LA, and CZ) independently conducted quality assessment of the studies included. After LA and CZ assessed the quality of 20% of the included articles (10% each) and FT assessed 20%, the results of the quality appraisal between LA and CZ (together) and FT were compared and found to be similar. To reduce duplication efforts, FT assessed the quality of all remaining studies. FT discussed any minor differences in the scoring of the remaining studies with LA and CZ as they arose. Studies with sample size < 50 were excluded. 

### 2.7. Data Extraction

Data were extracted using a data extraction proforma adapted from the Joanna Briggs Institute (JBI) [43] and other literature [44]. The data extraction proforma was pilot tested to ensure the relevance and uniformity of the data extracted. FT and LA conducted the pilot testing on three purposively selected papers, which led to the proforma being revised to better suit the NCD study context. 

The data extracted included the following: author and year, study aims, study design, study population, sample size and sampling technique, data collection and analysis methodologies, outcomes (prevalence of NCD risk factors), key findings (demographic characteristics, the prevalence of NCDs and their risk factors), author’s conclusion, and study limitations.

### 2.8. Data Analysis and Synthesis 

Studies that reported on NCD risk factors were summarised by type of NCD risk factor (i.e., alcohol consumption, smoking, and overweight and obesity). A meta-analysis of the effect size (prevalence) of various NCD risk factors was conducted using the Metaprop command in Stata 16 [45]. Studies that used different survey methods (i.e., heterogenous measurements of NCD risk factors) were narratively synthesised. Metaprop uses a binomial distribution to model the within-study variabilities and Freeman–Tukey double arcsine transformation to stabilize the variances [46]. 

Confidence intervals were based on exact binomial procedures in the Metaprop package. Statistical heterogeneity was assessed using the I2 measure [45] as an alternative to the test of heterogeneity (Q statistic). Both are susceptible to a small number of studies (*n* < 5); however, the test for heterogeneity was considered when *n* ≥ 10 (significance level at *p* < 0.05). 

All meta-analyses were performed using random-effects models even when tau2 (between-study variance) was zero, resulting in a similar effect size to a fixed-effects model. Sub-group analyses were conducted to explore potential between-study variation by region where the value of I^2^ was >50–75% (moderate to considerable) and the number of studies included was ≥10. The pooled prevalence of various NCD risk factors with associated 95% confidence intervals (CI) were examined visually using forest plots. Because of the small number of eligible studies (*n* < 10) for most of the NCD risk factors included, publication bias was not assessed (it is not informative in a meta-analysis with less than 10 studies) [47]. 

## 3. Results 

### 3.1. Description of Study Characteristics

This systematic review has been reported according to the Preferred Reporting Items for Systematic reviews and Meta-Analysis (PRISMA) [48]. Figure 1 presents the PRISMA flow diagram outlining the selection process of included studies. A total of 8434 records were retrieved from academic databases and grey literature sources. After duplicates were removed, 4731 records were screened by title and abstract against the eligibility criteria. Of these, 171 articles were selected for further full-text screening, and 47 fulfilled the inclusion criteria. At this stage, the major reasons for exclusion were ineligible population groups (e.g., children, pregnant mothers, people living with HIV, people with other comorbidities), wrong outcomes (e.g., studies that did not report on the prevalence of the risk factors, risk factors associated with glycaemic control, hypertension treatment practices, etc.), and ineligible study design. 

There were 376,162 participants in total in the studies included in the systematic review, and most were also included in the meta-analysis (*n* = 36). The largest sample size in a single study was 67,397 [49], whilst the smallest was 68 [50].

Six studies reported on the prevalence of multiple NCD risk factors [24,49,50,51,52], while the remaining examined only a single risk factor. All included studies employed descriptive and analytical cross-sectional study designs. Most studies were conducted in Amhara (*n* = 12), Oromia (*n* = 9), the Southern Nation nationalities and peoples’ region (SNNP) (*n* = 8), or nationally (Ethiopia) (*n*= 9). The remaining were conducted in Tigray (*n* = 3), Afar (*n* = 2), Dire Dawa (*n* = 2), Addis Ababa (*n* = 1), and Somali (*n* = 1). While Amhara, Oromia, and Tigray SNNP are relatively developed regions, the rest are less developed and more rural regions, typically characterized as hard-to-reach areas. Addis Ababa and Dire Dawa are the most urbanized areas in Ethiopia.

### 3.2. Methodological Quality Appraisal 

The quality assessment scores ranged from 4 to 10 (Appendix A). Twenty-nine studies scored 10 out of 10 [24,29,49,51,53,54,55,56,57,58,59,60,61,62,63,64,65,66,67,68,69,70,71,72,73,74,75,76], four studies scored 9 out of 10 [77,78,79,80], four studies scored 8 out of 10 [81,82,83,84], four studies scored 7 out of 10 [52,85,86,87], four studies scored 6 out of 10 [88,89,90,91], and two studies scored 4 out of 10 [50,52]. The major quality issues for studies that scored below 7 were not reporting non-response, the use of non-validated data collection tools, and not controlling for confounding.

### 3.3. NCD Risk Factors Combined 

Only one study reported on the combined prevalence of NCD risk factors from the Oromia regional state, finding that 70.9% of the study participants had at least one NCD risk factor [52]. 

### 3.4. Hypertension 

Twenty-seven studies measured the prevalence of hypertension (sample size ranged from 68 [49] to 67,397 [50]) (Appendix A). Two studies were conducted nationally in Ethiopia, with the remaining twenty-five studies conducted at regional state level. Meta analyses are presented both with and without the national data in Appendix A. 

I^2^ for the pooled prevalence of hypertension was 99.76, with *p* < 0.001 (Appendix A). To address this, we conducted sub-group analyses by regional states (Figure 2); however, the heterogeneity for the sub-group analysis remained high, likely due to the sub-groups (i.e., regional states) including fewer than 10 studies.

The overall pooled prevalence of hypertension including the national-level data was 21% (95% CI 17–27%) (the effect size (ES) in the forest plot represents the proportion, and it is these proportions converted into percentages that are reported from herein), and the prevalence of hypertension with the national-level data removed was 21% (95% CI 17–26%) (Figure 2). Across regions, the highest reported prevalence was in Oromia at 27% (95% CI 12–41%), and the lowest was in Addis Ababa at 17% (95% CI 15–19%) (Figure 2).

The two studies that were conducted nationally were pooled separately (Figure 3). The pooled prevalence of hypertension from these two studies was 15% (95% CI 15–16%). 

### 3.5. Overweight 

Seven studies reported on the magnitude of overweight [50,51,54,57,75,82,88] (Figure 4 and Appendix A). The pooled prevalence of overweight was 20% (95% CI 14.0−25%). While the highest magnitude of overweight was reported to be 29% (95% CI 26−32%) in SNNP, the lowest was in Oromia regional state at 9% (95% CI 6.5−11.4%). The heterogeneity between groups was I^2^ = 97.67%.

### 3.6. Obesity 

Seven studies reported on the prevalence of obesity [50,51,54,57,75,82] (Appendix A). The overall prevalence of obesity was 10.3% (95% CI 3.9–19.2%). Both the highest and lowest magnitudes of obesity were reported in Amhara regional state (Figure 5). The heterogeneity between groups was I^2^ = 98.63. 

### 3.7. Overweight and Obesity Combined 

One study conducted in Oromia state among adults reported on overweight and obesity, with a combined prevalence of 11.4% [52]. 

### 3.8. Physical Inactivity 

Five studies measuring physical activity levels in Ethiopia were identified, with sample sizes ranging from 548 to 10,260. All studies were conducted among men and women aged 15–69 years. In two studies conducted using the WHO STEPS survey in 2015, 94.2% of participants practiced physical activity as per the WHO recommendations [76]. In contrast, 66% and 83.1% of study participants reported some form of physical activity in two other studies conducted from 2008 to 2009 [87] and in 2019 [66], respectively. The first study was conducted in an urban centre in Dire Dawa [66], and the second was conducted in a demographic and health surveillance site (Gilgel Gibe) which comprised both urban and rural areas [87]. Another study conducted from 2008 to 2009 found that 18% of participants (*n* = 548) were physically inactive [88]. This study was conducted in an urban environment in the Afar regional state. 

### 3.9. Smoking 

Six studies conducted from 2012 to 2017 reported highly variable smoking prevalence, but also utilised different methods to measure smoking [61,74,86,87,88,92] (Appendix A). A study based on the 2011 demographic and health survey data reported a smoking prevalence of 3.21% [61], whilst another study conducted from 2008 to 2009 in southwestern Ethiopia (Oromia region) reported a prevalence of 9.3% [87]. A further study conducted in north-eastern Ethiopia (Afar region) from 2008 to 2009 reported a 13% prevalence [88]. A much higher smoking rate (29.5%) was found by Eticha and Kidane in northern Ethiopia (Tigray region) in 2013 [86]. However, the highest prevalence of smoking was 38.6% among males and 0.8% among females in a rural community in eastern Ethiopia in 2010 [74]. In a later study in north-western Ethiopia (Amhara region) conducted between 2009 to April 2015, 99.5% of participants reported they had never smoked [49]. Moreover, the prevalence of second-hand smoking in Southern Ethiopia (SNNP) in 2014 was 1% [68].

Two additional studies reported on the prevalence of various forms of smokeless tobacco use [65,93]. Lakew and Haile combined and analysed the 2011 and 2016 Ethiopian demographic and health survey data, and the prevalence of all forms of tobacco use was 4.1% [65]. In south-eastern Ethiopia (Oromia region), the prevalence of smokeless tobacco in 2015 among pastoralist communities was much higher at 45.3% [93]. 

### 3.10. Alcohol 

Alcohol consumption levels were reported in seven studies, with sample sizes ranging from 400 to 67,397 [49,60,70,72,87,88,91] (Appendix A). The national prevalence of alcohol drinking ranged from 13.5% in 2008–2009 [88] to 12.4% in 2015 [60]. Alcohol consumption was found to be more of a concern in regional studies, which reported the prevalence of harmful alcohol use ranging from 24.6% in 2013 [70] to 21.1% in 2014 [72]. 

In a study conducted in north-western Ethiopia (Amhara) from 2009 to 2015, 88.2% of participants reported drinking alcohol occasionally while 2.9% reported frequent alcohol consumption. However, the authors did not provide their definitions for occasional and frequent consumption [49]. In a study conducted in southern Ethiopia (SNNP) in 2013 [91], 65.4% of people reported consuming alcohol, while the prevalence of frequent alcohol use in southwestern Ethiopia (Oromia) in 2008–2009 was reported to be 7.1% [87]. 

### 3.11. Metabolic Syndrome and Cholestrol Level 

The studies that measured the prevalence of metabolic syndrome, triglycerides, and cholesterol used sample sizes ranging from 325 to 10,260 (Appendix A). 

Three studies reported directly on the prevalence of metabolic syndrome [24,27,78], with one (a national study conducted in 2015) reporting a prevalence of 4.8% [24], and another study conducted in 2017 in Oromia regional state reporting 9.6% [78]. The prevalence of metabolic syndrome was much higher (20.3%) in another study conducted in Addis Ababa [94]. 

Three studies assessed cholesterol levels [24,52,87]. The prevalence of high cholesterol in Oromia ranged from 5.5% in 2010 [52] to 10.7% from an earlier study in 2008–2009 [87]. The prevalence of high cholesterol level was found to be 5.2% in a later study conducted in Ethiopia in 2015 [24]. 

The prevalence of high blood triglycerides was also reported in one of these studies, found to be 7.7% in 2008–2009 [87]. 

### 3.12. NCD and Fruit and Vegetables 

Two studies reported on the prevalence of fruit and vegetable consumption, with sample sizes of 325 and 10,260 [52,88] (Appendix A). One study conducted in Oromia reported that the frequency of fruit and vegetable consumption in 2010 (amounts unmeasured) was in line with the recommended daily consumption [52]. On the other hand, a study in north-eastern Ethiopia (Afar) in 2008–2009 found that 97.9% of participants consumed less than the recommended amounts of fruits and vegetables [88]. 

## 4. Discussion

This systematic review and meta-analysis presents an updated evidence synthesis of the relative prevalence of NCD risk factors in Ethiopia, finding hypertension, overweight, and obesity to be the most prevalent. This review both updates the 2012 risk factor review for Ethiopia and allows an assessment of trends in risk factor prevalence in the ensuing 10 years. No meta-analysis was done in the earlier review, but there has been a substantial increase in the prevalence of hypertension, overweight, and obesity since 2011 from comparing the individual studies [28]. 

In the current meta-analysis and systematic review, the pooled prevalence of hypertension was 21%, with a substantial difference in the prevalence of hypertension between regional states in Ethiopia. That is, the highest prevalence of hypertension was in the Oromia regional state (27%), whilst the lowest was in Addis Ababa (17%). In the previous systematic review conducted in 2011 in Ethiopia, the prevalence of hypertension was also variable, ranging from 4.1% to 30% in different parts of Addis Ababa and being much lower (1.8%) in the rural region of Amhara [28]. 

A comparable overall prevalence of 29% was reported in a systematic review from Nigeria [95]. Another recent systematic review from Ethiopia reported wide variation in the magnitude of hypertension, ranging from 9.3%−30.3% in community-based studies, 7–37% in institution-based studies, and 13.2%–18.8% in a hospital-based study [96]. Two additional reviews found that the magnitude of hypertension was much higher in older Africans (47–55%) [97,98]. 

Overweight and obesity are also leading contributors to multiple NCDs. Globally, 2.8 million deaths are related to overweight and obesity [99,100], and the age-standardized prevalence of being overweight has been increasing since 2008 [101]. According to our review and meta-analysis, the overall prevalence of overweight was 19.2%, while the prevalence of obesity was 10.3% in Ethiopia, which is comparable with findings from other studies in sub-Saharan Africa that have reported a 21.6% prevalence in the region [102]. The prevalence of overweight was also 22.8% in Ghana [103] and slightly higher at 26.0% in Cameroon [104], where a 15.1% prevalence of obesity has also been estimated [104], which is slightly higher than the findings in the current study. The combined prevalence of overweight and obesity in this review was 26.8%, which is notably lower than findings from studies in Ghana (43%) [105] and Mozambique (30.5%) [106]. Together, this literature indicates that overweight and obesity are significant health burdens in Ethiopia and sub-Saharan Africa. 

With respect to physical activity, the current review found highly variable results in the proportion of people meeting the WHO-recommended levels, ranging from 18.1% [88] and 94.2% [76] among the Ethiopians sampled. Comparable magnitudes of WHO-recommended physical activity levels have been reported in studies in Uganda (94.3%) [107], Tanzania (96%) [108], and Nigeria (76%) [109].

The WHO-recommended level of physical activity for adults (aged 18–64) involves 150–300 min of moderate-intensity aerobic physical activity, or at least 75–150 min of vigorous-intensity aerobic physical activity, or an equivalent combination of moderate- and vigorous-intensity physical activity throughout the week [110]. The likelihood of meeting this will be related to occupational activity and subsistence, both of which are highly likely to vary across regional states in Ethiopia. 

The prevalence of smoking in the current review also varied by regional state, age, and other socioeconomic and demographic characteristics, but overall was relatively low compared with other sub-Saharan countries. The national prevalence of smoking in Ethiopia was 3.2%, while the prevalence in Oromia was 9.3% [87]. It is likely that the study in Oromia reported the highest prevalence of smoking [74] because smoking and Khat chewing is considered an important cultural ritual in these parts of Ethiopia [74]. Mixed findings were also reported in a study that involved several countries in sub-Saharan Africa, whereby cigarette smoking ranged from 1.8% in Zambia to 25.8% in Sierra Leone [111]. Nevertheless, evidence suggests that the prevalence of smoking is higher in multiple sub-Saharan African countries, including Sierra Leone (37.7%), Lesotho (34.1%), and Madagascar (28.5%), compared to Ethiopia (<10%) [112]. 

Globally, around 2.3 billion people currently consume alcohol [113]. In the current review in Ethiopia, the national prevalence of alcohol drinking was reasonably consistent, ranging from 12.4% to 13.5% [60], as was the prevalence of heavy drinking, ranging from 21.1% [72] to 24.6% [70]. In a previous systematic review in Ethiopia, the prevalence of current and lifetime alcohol use was 23.9% to 44.2%, respectively [114]. It remains difficult to compare the prevalence of alcohol consumption given heterogeneity in how alcohol consumption is measured (i.e., current drinker, episodic drinker, and lifetime drinker). 

Global evidence indicates that the increase in metabolic syndrome and population aging will continue to drive NCDs [115]. The prevalence of metabolic syndrome in the current review ranged from 4.8% [24] to 9.6% [78], which is much lower than the previously reported prevalence among women in sub-Saharan Africa (42.1%) [116]. Similarly, the prevalence of metabolic syndrome in Kenya (34.6%) [117] and South Africa (23.1%) [118] has been found to be notably higher than the prevalence in the current review. Country specific differences in behavioural, environmental, demographic, socioeconomic factors, and data quality may explain these differences. 

In the current systematic review and meta-analysis, several factors including being male, smoking, a family history, overweight, obesity, additional salt use, Khat chewing, and being married were associated with a higher prevalence of hypertension. Other factors such as high socioeconomic status, being married, and low physical activity were also found to be associated with overweight and obesity, as consistent with other studies from sub-Saharan Africa [119,120,121]. Similarly, being male, previous smoking, illicit drug use, and being of higher socioeconomic status were associated with smoking. The studies that we reviewed also found that being male, living in rural areas, being aged below 60 years, and having more life events and severe psychological distress were also associated with episodic drinking, which is consistent with the findings of other studies [122,123]. These factors that have been associated with NCD risk factors (i.e., the causes of the causes) were not the focus of this review, but we recommend that a robust systematic review and meta-analysis of these upstream influences on NCDs be conducted. 

### 4.1. Strengths and Limitations

The key strength of this study is that it is the first to comprehensive review and document the relative prevalence of various NCD risk factors in Ethiopia. Nevertheless, with the exception of hypertension and overweight, meta-analyses were not possible for most NCD risk factors because of heterogeneity and/or the small number of studies. Limitations exhibited from the primary studies such as social desirability bias and misclassification due to self-reporting should also be noted, and the small number of studies for some risk factors meant that the likelihood of publication bias could not be assessed. 

It should be noted that the studies included in this review are not representative of the entire nation of Ethiopia, as there is little or no published research on NCDs or NCD risk factors in Benishangul Gumuz, Gambela, and Harari regional states. Despite this, our review now highlights key gaps in NCD monitoring that need to be addressed by future research. 

### 4.2. Conclusions and Recommendation

This systematic review suggests that hypertension and overweight are the most prevalent NCD risk factors in Ethiopia, which was the case in the previous review as well. Given how common these risk factors now are, affecting approximately one-fifth of Ethiopians, the Ethiopian Government needs to invest in national NCD and NCD risk factor surveillance, with support from multilateral organisations, to further map and track these risks. Ethiopia has achieved a lot in terms of controlling and preventing HIV/ADIS, in collaboration with multilateral and bilateral organizations. Similar approaches could be employed to prevent and control NCDs by actively addressing the rise in risk factors. The ongoing civil war in various parts of the country will have major negative impacts on disease surveillance and prevention, and will further compromise the control of NCD risk factors and other health programs. Further research is needed to understand the relevance of tailored interventions, policies, and investments to prevent and address NCD risk factors in the Ethiopian context. 

## Figures and Tables

**Figure 1 ijerph-19-05316-f001:**
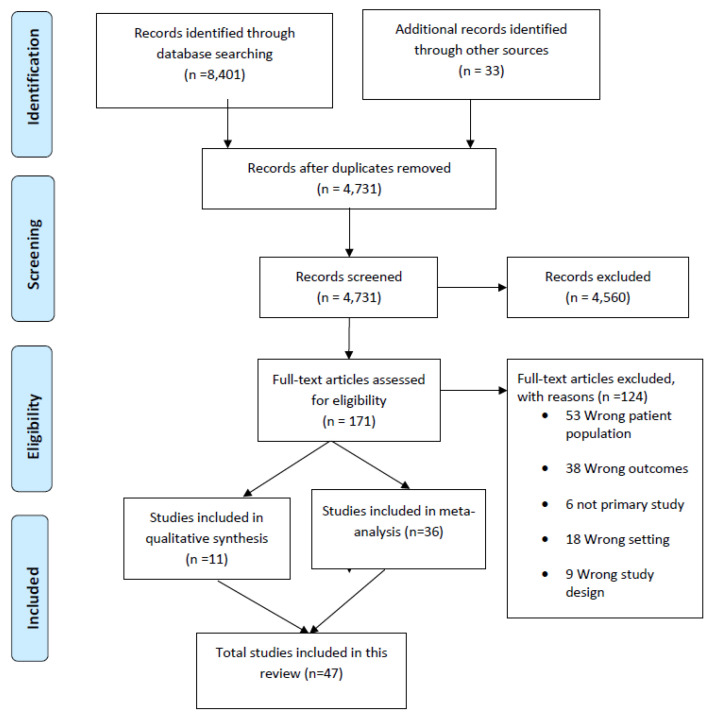
PRISMA diagram.

**Figure 2 ijerph-19-05316-f002:**
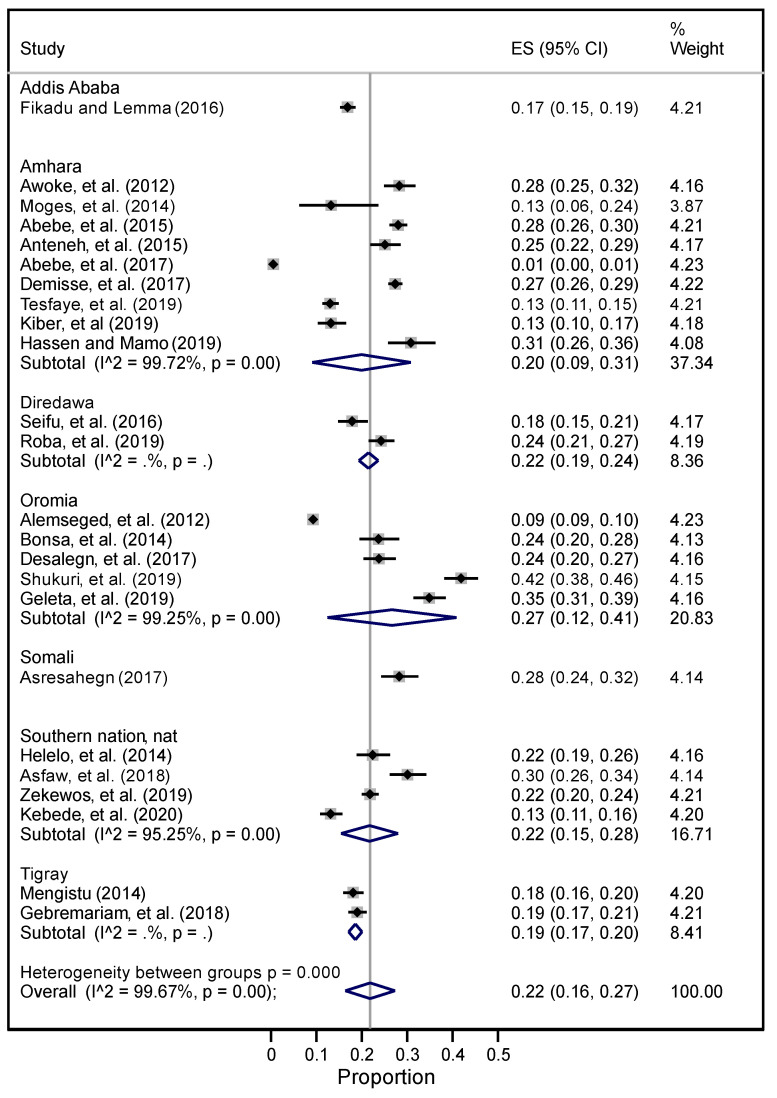
The pooled prevalence of hypertension among studies conducted at the national level in Ethiopia.

**Figure 3 ijerph-19-05316-f003:**
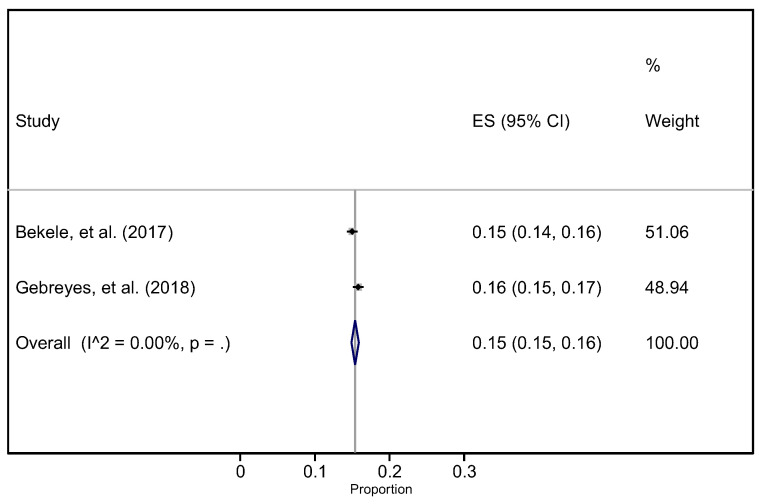
The pooled prevalence of hypertension by regional states in Ethiopia. **Note:** The large purple diamond highlights the sub-group pooled prevalence.

**Figure 4 ijerph-19-05316-f004:**
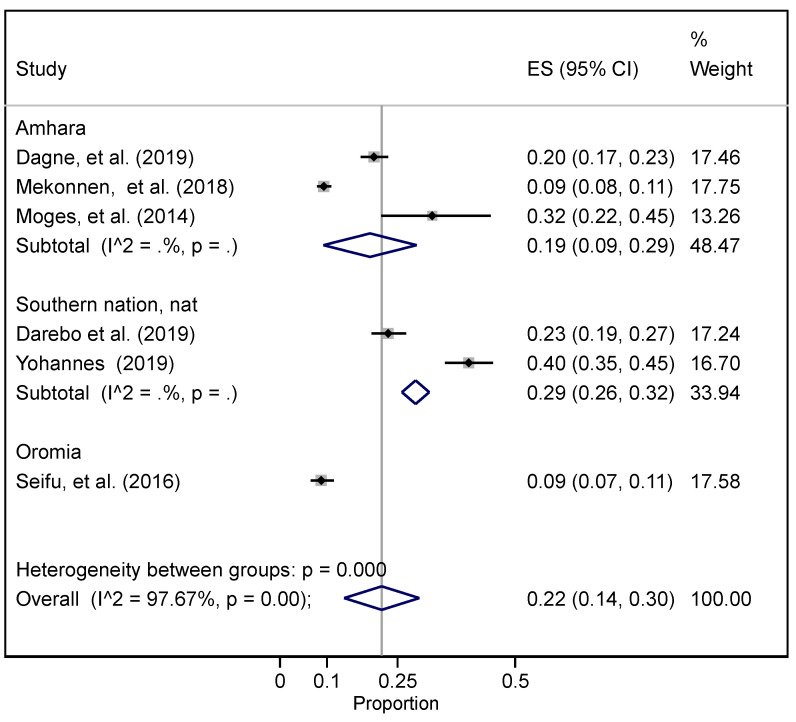
The pooled national and regional prevalence of overweight in Ethiopia.

**Figure 5 ijerph-19-05316-f005:**
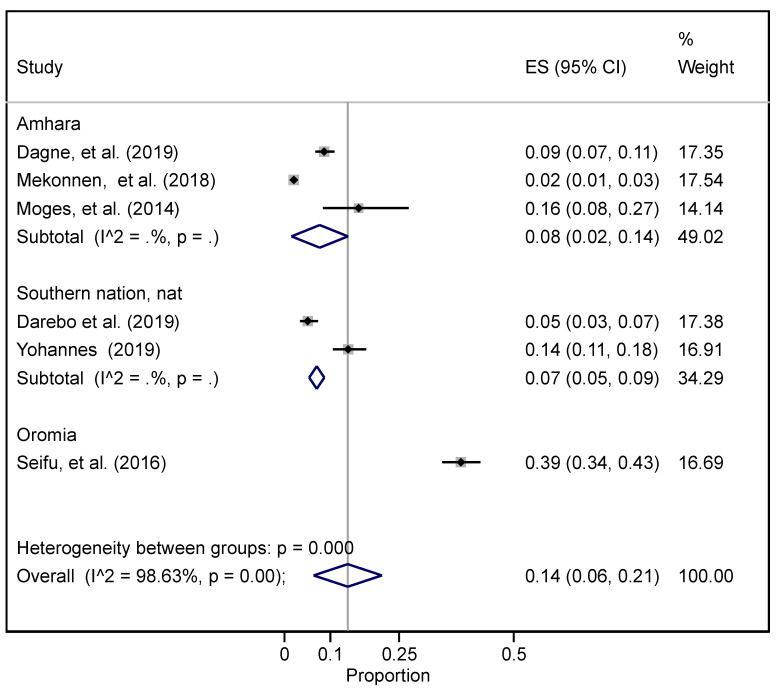
The overall and regional pooled prevalence of obesity in Ethiopia.

**Table 1 ijerph-19-05316-t001:** Key concepts and search terms.

Hedge 1: Indicators	Hedge 2: NCD Risk Factors	Hedge 3: Ethiopia
Prevalence* OR Proportion* OR Magnitude* OR Epidemiology* OR Pattern* OR Trend* OR Burden	Risk factors OR Hypertension OR (high blood pressure [MeSH Terms]) OR OR (hypertension [MeSH Terms]) OR hyperten* OR (hypertension [MeSH Terms]) OR Alcohol* OR (alcohol drinking [MeSH Terms]) OR Smoking* or (smoking [MeSH Terms]) or smok* or Tobacco or (tobacco [MeSH Terms]) OR Dietary* OR Food habit OR (food habit [MeSH Terms]) OR Unhealthy diet OR (eating behaviors [MeSH Terms]) OR Metabolic disorders OR Overweight OR (overweight [MeSH Terms]) or obes* OR (obesity [MeSH Terms]) OR Sedentary lifestyle* OR exercise OR (physical exercise [MeSH Terms]) OR physical activ* OR Physical activity OR Cholesterol or (cholesterol [MeSH Terms])	Ethiopia OR Tigray OR Amhara OR Oromia OR Afar OR Somali OR Gambela OR Benshangul Gumuz OR Addis Ababa OR Diredawa

## Data Availability

Availability of raw data is not applicable in this study, but we are happy to provide what we extracted from the included papers.

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
