# Peer review of "The Magnitude of NCD Risk Factors in Ethiopia: Meta-Analysis and Systematic Review of Evidence"

_ijerph, 2022, doi:10.3390/ijerph19095316_

Round 1

Reviewer 1 Report

This is an important manuscript, particularly considering the rising prevalence of NCDs in sub-Saharan Africa. However, the authors can strengthen the quality of their work by improving its clarity and by proofreading. Additionally, the writing can be more succinct and more refined.

Abstract

There appears to be an extra space between the colon after the word, recommendations, and before the word, Approximately.

Introduction

  • The authors repeatedly mention that NCDs are not a priority for many governments. I would urge them to also consider why this is the case instead of simply stating that it is not a priority.
  • Page 2, line 53: Consider including hyphens for the expression, low and middle income countries and introduce the abbreviation, LMICs, here rather than later on in the paragraph.
  • Page 2, line 69: There is an extra space in the sentence.
  • Can the authors include some information on the prevalence and incidence of NCDs in sub-Saharan Africa as well as in Ethiopia?
  • Can the authors briefly state some cultural factors associated with NCD risk factors in sub-Saharan Africa?
  • Can the authors provide some more justification for the conducting a systematic review/meta-analysis? In African countries where these reviews/analyses have already been conducted, to what extent have those facilitated health policies or public health efforts?

Methods

  • For the context/setting, the authors should consider including a few sentences about health access/health delivery in Ethiopia and the existing IDSR or PHEM systems for disease surveillance.
  • If possible, can the authors include the MEDLINE search terms in an appendix?
  • Page 5, line 194: What are one or two examples of minor uncertainties?
  • Page 5, line 202: Earlier in the manuscript, the authors mentioned that they excluded studies conducted in a health facility. Why do the setting options for data extraction then include “health facility” if that is indeed what they did?
  • Page 5, line 206: Can the authors provide more details on what they mean when they say other relevant information?
  • Page 5, line 209: The following sentence is unclear, “Primary studies that reported on NCD risk factors were summarised by type.” I am not sure what type If the authors mean that studies were summarized based on the type of NCD risk factors, that needs to be clearer. Also, what are the authors referring to when they say primary studies?
  • Page 5, lines 212-215: Can the authors make the following sentence clearer as it is difficult to follow what they did? “The Metaprop command requires two variables in the format {n, N} such that prevalence =n/N is declared and then uses a binomial distribution to model the within-study variabilities or Freeman-Tukey double arcsine transformation to stabilize the variances”
  • Page 6, line 221: I think the authors intended to say performed, not preformed.

Results

  • Page 11, line 310: The authors should consider saying, data from the 2015 Ethiopian STEPS survey, instead of the current phrase.
  • Page 12, line 318: I think the authors intended to say due to differences, not due differences.
  • Page 12, lines 343-345: The authors need to cite the additional study that they are referencing in the following sentence, “In a study conducted in north-western Ethiopia (Amhara), 88.2% of participants reported drinking alcohol occasionally while 2.9% reported frequent consumption in a study conducted in 2009 to 2015 [42].” Currently, they have cited only one study, and the citation does not match the preceding phrase.
  • Page 12, lines 343-347: Can the authors briefly indicate what is meant by drinking alcohol occasionally and frequent alcohol use?
  • Page 13, lines 365-366: How is the prevalence of fruit and vegetable consumption universal? Can the authors be more specific?
  • Page 13, lines 367-368: How are the authors defining adequate levels of fruit and vegetable consumption?

Discussion

  • Page 13, line 400: Can the authors briefly state the WHO recommendation for physical activity?
  • Page 14, lines 431-433: Can the author refine the sentence below? It is repetitive to say that risk factors for hypertension are associated with hypertension.

“In Malawi, risk factors for hypertension such as smoking, unhealthy diet, excessive alcohol drinking, and physical inactivity were associated with hypertension [117].”

  • It may be important for the authors to consider and include a few other limitations. For instance, the studies included in their review/analyses are not representative of the entire country as there is little research/ no published research on NCDs or NCD risk factors in Benishangul, Gambela, etc.
  • Can the authors explain why support from international organizations is necessary for NCD/NCD risk factor surveillance in Ethiopia?
  • Does the current state of Ethiopia have any implications for the adequate and timely surveillance of diseases, including NCDs and related risk factors? The authors should consider including one or two sentences on this issue.

Author Response

We have attached it in a table format. 

Reviewer 2 Report

The authors provided an updated systematic review and meta-analysis describing prevalence of risk factors for various non-communicable diseases in Ethiopia. Several relevant risk factors were selected, and it was estimated that a substantial proportion of Ethiopia’s population is at risk for NCDs based on these risk factors. The review provides insight into multiple health conditions in a populous nation.

MAJOR COMMENTS

  1. Noncommunicable diseases are described as the motivation of this work, but they are not explicitly defined. Is there a specific list of NCDs of interest from which the authors developed the list of risk factors? This would also help clarify why the particular risk factors were selected.
  2. The I2 statistic was used for heterogeneity, but is never reported. The authors should report this. It would also be beneficial to discuss the heterogeneity (or lack thereof) across studies and the implications of this for the final results and conclusions.

MINOR COMMENTS

  1. In Table 1, it is unclear why the middle column has the word “OR” bolded in some instances but not others. The third column also contains an unnecessary “OR” at the end.
  2. Lines 169-171: For inclusion, did the studies have to report prevalence of risk factors as their primary aim, or were other primary objectives permitted for inclusion?
  3. Lines 211-212: Elaborate on heterogeneous measurement. Did this refer to definitions of these risk factors or differences in survey methods across studies?

Author Response

Response to reviewer 2 has bee attached in a table format below 

Round 2

Reviewer 1 Report

I appreciate the authors’ clear revisions to the paper. However, there are still some issues with punctuation, proofreading, etc. I highlighted some of those issues below, and I recommend that the authors consider proofreading or using a professional proofreading service to strengthen their work.

Introduction

  • Page 2, line 258: a period is missing after the citations (11-14).
  • Page 2, link 268: there is an extra space between the words, have and
  • Page 2, line 283: The authors should consider removing the s in Since that sentence is now including about both globalization and urbanization, the authors should also consider saying these factors concurrently increase instead of it currently increases.
  • Page 2, line 289: What does xxxx stand for? I assume the authors forgot to include the specific years/dates.
  • Page 2, line 300: The authors should delete the word in that comes before 2015.

Methodology

  • Page 3, line 565: Since the authors actually list all ten regions, they can say These are instead of These include, especially since they use the word include again later in the sentence.
  • Page 3, lines 579: A period is missing after the references provided.
  • Page 5, lines 896-897: The authors need to edit or delete this statement.
  • Page 7, line 1213: what are d random effects models?

Results

  • Page 7,line 1236: The authors should indicate the actual n instead of using xxxx. I assume this was an oversight.
  • Page 7, line 1244: I am surprised that the authors described Addis Ababa and Dire Dawa as rural regions when these are actually two major, urban cities in Ethiopia. Again, I assume this was an error.
  • Page 12, line 1528: The authors should consider removing the question mark or confirming if what they are reporting is indeed pooled prevalence.
  • Page 13, line 1592: There seems to be some missing information in this statement.

Discussion

  • Page 17, line 2665: The authors should provide the right information at the end of the sentence with regards to trends during the last review.

Author Response

We have attached the point by point response to the reviewers feedback.
